

# Long-term changes in Central European river discharge 1869-2016: impact of changing snow covers, reservoir constructions and an intensified hydrological cycle

Erwin Rottler[1], Till Francke[1], Gerd Bürger[1], and Axel Bronstert[1]

[1]Institute of Environmental Science and Geography, University of Potsdam, Karl-Liebknecht-Straße 24–25, 14476 Potsdam, Germany

**Correspondence:** Erwin Rottler (rottler@uni-potsdam.de)

**Abstract.** Recent climatic changes have the potential to severely alter river runoff, particularly in snow-dominated river basins. Effects of changing snow covers superimpose with changes in precipitation and anthropogenic modifications of the watershed and river network. In the attempt to identify and disentangle long-term effects of different mechanisms, we employ a set of analytical tools to extract long-term changes in river runoff in high resolution. We combine quantile sampling with moving

average trend statistics and empirical mode decomposition and apply these tools to discharge data recorded along rivers with nival, pluvial and mixed flow regimes as well as temperature and precipitation data covering the time frame 1869-2016. With a focus on Central Europe, we analyze the long-term impact of snow cover and precipitation changes along with their interaction with reservoir constructions.

Our results show that runoff seasonality of snow-dominated rivers decreases. Runoff increases in winter and spring, while

discharge decreases in summer and beginning of autumn. We attribute this redistribution of annual flow mainly to reservoir constructions in the alpine ridge. During the course of the last century, large fractions of the alpine rivers have been dammed to produce hydropower. In recent decades, runoff changes induced by reservoir constructions seem to overlap with changes in snow cover. We suggest that alpine signals propagate downstream and affect runoff far outside the alpine area in river segments with mixed flow regimes. Furthermore, our results hint at more (intense) rainfall in recent decades. Detected increases in high

discharge can be traced back to corresponding changes in precipitation.

## 1 Introduction

In many regions of the world, rivers constitute essential lifelines and form the basis of human livelihood. However, recent climate changes may severely affect the hydrological cycle and jeopardize the functional diversity of river systems. Most severe changes are expected to occur in snow-dominated river basins. In a warmer world, snow cover characteristics and snow

melt contribution to river runoff will change fundamentally. Rising temperatures are expected to cause less winter precipitation to fall as snow and existing snow covers to melt earlier in spring (Barnett et al., 2005; Simpkins, 2018; Kormann et al., 2015; Birsan et al., 2005). Recent studies suggest that rainfall amount and the number of extreme rainfall events increase due to warmer air holding more water along with enhanced evaporation (Lehmann et al., 2015; Coumou and Rahmstorf, 2012;





Mueller and Pfister, 2011). Investigating changes in features of snowpack and snowmelt for key mountain regions, Stewart (2009) summarizes "that both temperature and precipitation increases to date have impacted mountain snowpacks" already. For the Rhine river, one of the most important rivers in Europe, Stahl et al. (2016) indicate that "the influence of climate change is visible particularly in the temporal shifts of seasonal minima and maxima of the hydrological regimes of snow and glacier
melt dominated alpine headwater catchments."

In addition to changes in snow covers and precipitation, anthropogenic modifications of land surface, subsurface properties and the river network alter river runoff. During the 20th century, more than 45 000 large dams were constructed around the world (Word Commision on Dams, 2000). Also in the Rhine river basin, human activities change runoff with regard to amount, its temporal distribution as well as water quality (Wildenhahn and Klaholz, 1996; Belz et al., 2007; Wildi et al., 2004).

The current knowledge on how climatic changes and changing watershed properties impact river runoff comes largely from instrumental records of hydro-climatic variables, particularly temperature, precipitation and runoff. Birsan et al. (2005) state that "as a spatially integrated variable streamflow is more appealing for detecting regional trends than point measurements of precipitation which is highly variable in space and time", but also point out that watershed properties and their changes over time constitute an "obvious complication in interpreting trends in streamflow data." In addition, quality and length of recorded
time series often is insufficient to identify and disentangle effects of the various mechanisms. A sufficient length of the time series inter alia is crucial to be able to distinguish between natural climate variability and signals of climate change. Variability of large-scale atmospheric flow on annual to multi-decadal scales, for example, can cause variations in hydro-climatic data, which can either counterbalance or reinforce signals of long-term changes (Hanson et al., 2006; Frei et al., 2000; Kerr, 2000; Scherrer et al., 2016). Studies preparing and investigating long time series of high quality are of great importance and form
the basis of our current understanding of features and magnitudes of recent climatic changes (e.g. Vincent et al., 2002; Begert et al., 2005; Schmidli and Frei, 2005; Moberg et al., 2006; Scherrer et al., 2016). However, to further consolidate and extend findings obtained so far, new sets of analytical tools to extract information stored in this time series need to be developed, tested and applied on climatological and hydrololgical records.

Our study aims at a better understanding of long-term changes in river runoff and identifying potential underlying driving
mechanisms, by analyzing daily resolution hydro-climatic time series recorded in Central Europe between 1869 and 2016. We assess long-term changes in a highly resolved manner by combining quantile sampling, moving average trend statistics and empirical mode decomposition. The two main research question we want to address are:

– What is the long-term impact of changes in snow cover on river runoff?

– How do runoff changes induced by changes in snow cover compare with changes caused by reservoir constructions and
changes in precipitation?





## 2 Study area and Data

We investigate discharge time series from four gauging stations (Fig.1 and Tab.1). The depicted gauges stand out by the exceptional length of their records and represent different types of flow regimes: nival, pluvial and complex. Gauge Wasserburg is located at the Inn river in Upper Bavaria, Germany. The Inn river is a right tributary of the Danube. The river´s source is located in the Swiss Alps and most of its drainage area ($1,20 \cdot 10^4$ km² until gauge Wasserburg) possesses high alpine character. The other three gauges investigated, namely Basel, Wuerzburg and Cologne, are located in the Rhine river basin. The Rhine river is one of the largest rivers in Europe. It is a heavily used waterway and livelihood for the region. At gauge Basel, river runoff is dominated by snowmelt and rainfall-runoff from the Alps. Gauge Wuerzburg is located at the Main river in northern Bavaria, Germany. The Main river is a right tributary of the Rhine river. The catchment area until gauge Wuerzburg is $1,40 \cdot 10^4$ km². The city of Cologne is the largest city along the Rhine river and located in the Lower Rhine region after the confluences with all major tributaries. Until Cologne, the Rhine river drains an area of $1,44 \cdot 10^5$ km². For all selected gauges, discharge data in daily resolution are available at least since 1869. For gauge Basel, statistical test show that measured discharge is homogeneous since 1869, i.e. values are free from anthropogenic effects such as change in instrumentation, change in daily recording frequency or lowering of the river bed (Pfister et al., 2006). Other gauging stations investigated are part of the hydrometric observation network of the water authorities in Germany. Recordings are regularly checked to ensure high quality and reliability. Discharge times series were obtained from The Global Runoff Data Centre (GRDC). Data from GRDC were used as-is without any further treatment. Elevation distributions and monthly Pardé-coefficients for investigated river basins are presented in the appendix (Fig.A1).

Furthermore, we analyze daily resolution temperature and precipitation data provided by the Federal Office of Meteorology and Climatology of Switzerland (MeteoSwiss). At MeteoSwiss, a standardized homogenization procedure is applied on a set of monthly temperature and precipitation time series (Begert et al., 2005). During this homogenization procedure attained monthly correction values also are applied on daily resolution data. The homogenization of long climatological time series is necessary to correct for non-climatic factors influencing the data. Currently, homogenized daily temperature/precipitation data are available for 28/73 stations. In the following, we focus on meteorological stations were both temperature and precipitation data are available at least since 1869 and there is no gap in the data longer than 60 days. In total, nine stations fulfil these criteria. Results of the three most prominent stations are displayed and discussed in the main manuscript (Fig.1 and Tab.1), information on (Tab.A1) and results of (Fig.D1) the remaining stations are given in the appendix.

## 3 Methods

To detect long-term changes in the investigated hydro-climatic data, we combine quantile sampling with moving average trend statistics and empirical mode decomposition (EMD). The selected analytical tools and their combined application to daily time series enable a highly resolved investigation of changes throughout the investigated time frame. The analysis is divided into four steps. Each analysis step complements and extends the information of the previous one, so that step by step, a comprehensive picture of long-term changes takes shape (Fig.3).



### 3.1 Seasonality of river runoff

To investigate the seasonality of river runoff, we estimate quantiles on a daily basis (QDAY). For every day of the year (DOY), we take all available measurements (i.e. 148 daily values for the period 1869-2016) and calculate QDAYs empirically for probabilities ranging from 0.01 to 0.99. In the framework of this study, quantiles are calculated as the $\frac{k-1/3}{n+1/3}$ plotting position, with n as the sample size and k = 1,....n being the rank (e.g. Hyndman and Fan, 1996). This approach corresponds to type 8 of the function 'quantile' in the R environment (R Core Team, 2018).

### 3.2 Changes in seasonality

In order to get a first insight into changes in runoff seasonality, we estimate quantiles within a 30 day moving window (QMOV). To assess the temporal evolution of these values over the observation period, we employ trend analysis. We calculate trend magnitudes of QMOV using the robust Theil-Sen trend estimator (TST) on a daily basis for all quantiles. Within the linear regression approach of TST, trend magnitudes are estimated as the median slope of ranked data values (Theil, 1950; Sen, 1968; Bronaugh and Werner, 2013).

### 3.3 Onset and evolution of changes

The use of linear trends to quantify the temporal evolution of hydro-climatic variables often lacks physical justification. The respective signals are likely to be non-linear (Fig.2 c and d). Even when using parametric functions for capturing the non-linear behaviour, e.g. exponential or power law functions, it is not guaranteed that results reflect the actual characteristics of underlying processes in the data. An adaptive approach, which does not require a predetermined basis function, is required to get a more flexible characterization of the trend. We employ EMD for this purpose. EMD is an empirical, direct and adaptive method to analyze non-linear trends. It decomposes the signal into oscillatory modes and provides a powerful tool to separate short time-scale signals from a general trend (Wu et al., 2007; Huang et al., 1998; Luukko et al., 2016; Huang et al., 1999). To avoid mode mixing issues, we performed EMD on an ensemble of the initial data signal: Ensemble EMD (EEMD) (Wu and Huang, 2009). Each ensemble member is perturbed by low-amplitude white noise and the results are averaged at the end of the computations. To keep the characteristics of a complete decomposition, i.e. all extracted intrinsic mode functions (IMF) sum up to the original signal, the averaging process is carried out separately for each IMF component (Torres et al., 2011). This extension results in a Complete EEMD with Additive Noise (CEEMDAN). We use an ensemble of 10000 members, a noise strength of 0.5 times the standard deviation of the input signal and the R package 'Rlibeemd' (Luukko et al., 2016) to perform CEEMDAN. The residual of CEEMDAN "can be used to represent the intrinsic trend of the data" (Luukko et al., 2016).

We assess these residuals for discharge, temperature and precipitation on a daily basis after calculating moving average values within a window with a width of 30 days for discharge and temperature and a width of 90 days for precipitation. To make results of different days comparable, we center each residual by subtracting its mean. To enable the comparison between CEEMDAN residuals and more commonly used linear approaches, we assess, whether the non-parametric Mann-Kendall





trend test (MK) detects statistical significant monotonic trends in the data CEEMDAN was applied on ($\alpha = 0.05$) (Mann, 1945; Kendall, 1975). Days with significant monotonic changes are marked with points on top of respective plot panels.

## 3.4 Changes in quantiles

Furthermore, we investigate changes in quantile magnitudes over time. Therefore, quantiles are estimated on an annual level
(QYEA) (Fig.3). The temporal evolution of QYEAs over the investigated time frame is assessed applying CEEMDAN. In the case of precipitation, we only use values from 'rainy days' (i.e. precipitation > 1 mm). The MK-test serves to assess the significance of the trends (marked with points on top of the panel (see section 3.3)).

## 4 Results

### 4.1 Seasonality of river runoff

Runoff recorded at gauges Wasserburg and Basel is highly seasonal with high/low runoff during summer/winter (nival flow regime) (Fig.4 a1, b1). Compared to gauge Wasserburg, more runoff is recorded at Basel during winter, i.e. the contrast between summer and winter is less pronounced. At gauge Wasserburg, very high discharge values are almost solely recorded between the months of May and September. Conversely, at gauge Basel, floods occur throughout the year. Downstream gauge Basel, the flow regime of the Rhine river evolves towards a complex regime (see gauge Cologne Fig.4 c1). Runoff from rain-dominated
tributaries such as Neckar, Main and Mosel blend with alpine runoff. Rainfall-runoff dominated basins are characterized by high discharge during winter and beginning of spring and low discharge in summer, as seen for Wuerzburg (Fig.4 d1).

### 4.2 Changes in seasonality

At gauges Wasserburg and Basel, runoff increases during winter and spring for all quantiles, while it decreases during summer and beginning of autumn (Fig.4 a2, b2). This corresponds to a reduction in runoff seasonality. A very similar overall pattern of
changes in runoff can be detected at gauge Cologne: runoff increases during winter and spring and decreases during summer and autumn (Fig.4 c2). In contrast, at gauge Wuerzburg, discharge quantiles increase throughout, except for high levels during end of February and March (Fig.4 d2). Similar to gauges Basel and Cologne, the strongest increases occur during winter.

### 4.3 Onset and evolution of changes

At gauge Wasserburg, pronounced changes in seasonality start in the second half of the 20th century during the 1960s (Fig.4
a3). In contrast, changes at gauge Basel seem to be more gradual and starting earlier in the investigated time period already (Fig.4 b3). At gauge Wuerzburg, a clear onset of change cannot be detected (Fig.4 d3), however, increases seem to be more uniform and enhanced in recent decades. Patterns of change from snowmelt and rainfall-runoff dominated tributaries overlap at gauge Cologne (Fig.4 c3).





Looking at the respective evolution of potential drivers, temperatures continuously increased throughout the year (Fig.5 a1, b1, c1). Similar amplitude and interannual patterns are apparent in the three time series. The amount of precipitation increases in recent decades, particularly during winter (Fig.5 a3, b3, c3). The MK trend test detects significant monotonic increases/decrease in runoff during winter/summer for gauges Wasserburg and Basel (Fig.4 a3, b3). For temperature, the MK

detects significant increases throughout the year (Fig.5 a1, b1, c1). Precipitation increases significantly during winter (Fig.5 a3, b3, c3).

### 4.4    Changes in quantiles

Since the 1960s, QYEAs strongly increase/decrease at levels below/above 0.6 at gauge Wasserburg (Fig.4 a4). These changes in QYEAs correspond to the strong decrease in seasonality in recent decades (see section 4.2): Runoff diminishes in summer

and increases in winter. Likewise, at gauge Basel lower QYEAs (levels < 0.6) increase and higher QYEAs (levels 0.6 - 0.8) decrease (Fig. 4 b4). However, the onset of changes is earlier and changes are smoother compared to detected signals at gauge Wasserburg. Particularly changes in low QYEAs start to increase at the beginning of the investigated time frame already. Contrary to results from gauge Wasserburg, QYEAs at highest levels (> 0.8) have been increasing at gauge Basel since the 1960s (Fig.4 b4). QYEAs from gauge Wuerzburg increase over the entire range investigated (Fig.4 d4). Changes in QYEAs

below a level of approximately 0.6 occur earlier and are smoother than for higher levels. There, the increases are enhanced in recent decades. At gauge Cologne, high QYEAs increase in recent decades (Fig.4 c4), making it similar to findings from gauge Basel and Wuerzburg. Also the lower QYEAs experience an increase. This increase, however, is not a gradual one over the entire time frame, but rather a U-shaped process (decline until the 1940s, then increase).

For precipitation, we detect a similar pattern. Increasing QYEAs hint at more (intense) rainfall in recent decades (Fig.5 a4,

b4, c4). Increases in QYEAs in temperature seem to occur earlier and seem to be enhanced at lower temperatures (Fig.5 a2, b2, c2). Changes in quantiles for individual seasons are given in the appendix (Fig.B1 and Fig.C1). At gauge Wasserburg, changes in QYEAs are significant according to the MK (Fig.4 a4). The more a trend pattern deviates from a monotonic increase and more U-shaped signals emerge, the more often the MK results in non-significant p-values. Main results depicted in Fig.4 are summarized in Tab.2.

## 5    Discussions

### 5.1    Seasonality of river runoff

Runoff at gauge Wasserburg is dominated by the accumulation and depletion of a seasonal snow cover. The intra-annual variability of runoff is very high and floods mainly occur during the snow melt season and during summer, when higher temperatures enable liquid precipitation in large fractions of the catchment (Fig.4 a1). There are no bigger lakes that could

attenuate flood or low flow events generated in the basin. In comparison, large lakes constitute an important element of the Rhine river basin until gauge Basel. Furthermore, large parts of the basin are sub-alpine terrain. As a result, liquid rainfall is





an important streamflow component throughout the year and runoff less seasonal compared to gauge Wasserburg (Stahl et al., 2016). Reconstructing the largest flood events in the High Rhine basin since 1268, Wetter et al. (2011) indicate that about half of all major floods occur during summer. Flood events during summer usually are the result of high baseflow due to a melting alpine snow cover superimposing with heavy rainfall (Wetter et al., 2011). Extreme flood events during autumn, winter

and spring often are caused by long-lasting precipitation events coinciding with strong snow melt due to rain-on-snow (RoS) and/or a temporary temperature increase (Wetter et al., 2011; Schmocker-Fackel and Naef, 2010). For higher elevated river basins, RoS events play an important role in runoff formation (Sui and Koehler, 2001; Merz and Blöschl, 2003). The RoS flood occurring in the Bernese Alps, Switzerland in October 2011 showed how damaging these kind of events can be (Rössler et al., 2014). Another example are the RoS events from January 2011, where rainfall released vast amounts of water stored in

a temporary snow cover and caused RoS-driven flood events in whole Central Europe (Freudiger et al., 2014).

Even though about one third of the runoff in the Main river originates from snow melt (Stahl et al., 2016), there is only little impact of snow accumulation and melt on the seasonal distribution of discharge. It seems that low temperatures rarely prevail long enough to enable the accumulation and preservation of snow over a longer period. Runoff is dominated by large-scale rainfall events occurring in winter and increased evapotranspiration during summer (Fig.4 d1). At gauge Cologne, we have the

situation of superimposing nival and pluvial runoff components (Fig. 4 c1). This overlap results in a more uniform seasonal distribution of discharge. High QDAYs are higher during winter, whereas low QDAYs are higher during summer (Fig.4 c1). This reversal in the seasonal distribution hints at the importance of different flow components for different flow situations. Runoff due to large-scale rainfall events over middle and lower parts of the catchment are important for high discharge values, particularly during winter. During summer, snow and glacier melt from the alpine part of the basin play an important role for

the sustenance of runoff in the lower reaches of the Rhine river (Stahl et al., 2016).

## 5.2   Changes in seasonality

In the snow-dominated river basins Wasserburg and Basel, the seasonality of river runoff decreases over the investigated time frame. For the increasing runoff values during winter and early spring, several mechanisms have to be taken into account. First of all, changes in the alpine snow cover have to be considered. In recent decades, rising temperatures cause less snow

accumulation during winter (Laternser and Schneebeli, 2003; Marty, 2008; Scherrer et al., 2004; Wielke et al., 2004). Thus, a greater fraction of total precipitation is liquid and reaches the river system without being stored in snow packs. In addition, the frequency of days with temperatures above 0 °C increases, causing parts of any existing snow cover to melt (Scheifinger et al., 2003; Kreyling and Henry, 2011; Zubler et al., 2014; Schädler and Weingartner, 2010). Rising temperatures also result in shorter snow duration, where "shorter snow duration is mainly caused by earlier snow melting in spring than by later first

snowfalls in autumn" (Laternser and Schneebeli, 2003). The earlier onset of snow melt in spring represents a much-noticed effect of rising temperatures on alpine river runoff (e.g., Kormann et al., 2015; Birsan et al., 2005; Stewart, 2009).

Less snow accumulation during the preceding winter results in lower discharges during the following melting period, i.e. late spring and early summer. Furthermore, recent studies suggest that rising temperatures might lead to a reduction in snow melt rates (Musselman et al., 2017; Wu et al., 2018). "Slower snowmelt in a warmer world may decrease the likelihood that





wetness thresholds that permit hydrologic connectivity will be exceeded, leading to spring and summer streamflow declines and lower runoff efficiency" (Musselman et al., 2017).

Changes in the liquid/solid fraction of precipitation overlap with changes in the total amount of rainfall. We observed increased rainfall during winter for all stations investigated. Likewise, numerous other studies point at a recent increase in

precipitation, particularly during winter (e.g., Begert et al., 2005; Scherrer et al., 2016; Frei and Schär, 2001). However, increasing catchment evaporation due to increasing radiation, air temperature and vegetation activity might at least partly compensate detected changes in precipitation (Duethmann and Blöschl, 2018; Schädler and Weingartner, 2010; Norris and Wild, 2007; Wild et al., 2007).

At gauge Cologne, we also detect a decrease in discharge during summer and autumn (Fig.4 c2). We hypothesize that this

decrease is the result of a downstream propagation of the alpine signal, possibly overlapping with increasing evaporation rates in the basin. The decrease in summer discharge in the Lower Rhine cannot be attributed to reduced ice melt contributions from the alpine glaciers. Assessing the snow and glacier melt components of streamflow of the Rhine river for the time frame 1901-2006, Stahl et al. (2016) showed that "despite the glacier retreat the modelled ice melt component of the streamflow in the Rhine does not show a strong long-term trend over the entire study period, i.e. a systematic decline or increase of this

component. The detailed results of the modelling suggests that an increased ice melt due to increased temperature may have been compensated by the reduction in glacier area". Gauge Wuerzburg, with its discharge increasing throughout the entire year, does not show any detectable changes in seasonality.

## 5.3   Onset and evolution of changes

Investigating long-term snow trends of the Swiss Alps, Laternser and Schneebeli (2003) suggest that "mean snow depth, the

duration of continuous snow cover and the number of snowfall days in the Swiss Alps all show very similar trends during the observation period 1933-99: a gradual increase until the early 1980s (with significant interruptions during the late 1950s and early 1970s) followed by a statistically significant decrease towards the end of the century".

At gauge Basel, these changes in snow cover seem to be insufficient to explain the decrease in runoff seasonality detected. Particularly winter discharge (low QYEAs) increases already from the beginning of the investigated time frame on (Fig.4

b3 and b4). Instead, we suspect anthropogenic alterations of the river network, particularly reservoir constructions, to be an important driver. These might have caused the redistribution of water from summer to winter earlier in the investigated time frame already. Large fractions of the Swiss and Austrian alpine river systems have been dammed to produce hydropower. The two alpine countries have the highest specific hydroelectric production per surface area globally (Truffer et al., 2001). The first hydropower station in Switzerland was constructed in 1899 (Verbunt et al., 2005). Dam constructions in the Alpine Rhine and

along other alpine rivers, such as Aare, Limmat and Reuss, gained momentum in the 1920s and most of the large storage lakes were constructed between 1950 and 1970 (Meile et al., 2011; Wildenhahn and Klaholz, 1996; Wagner et al., 2015) (Fig.E1). The total storage volume of large storage lakes (river weirs not included) of the High Rhine/entire Rhine basin is estimated to amount to $1.86/3.12 \cdot 10^9$ m³ (Wildenhahn and Klaholz, 1996). In order to ensure full functional capability of high-head storage hydropower stations, reservoirs need to have sufficient water volume stored at all times. Therefore, reservoirs tend to





be filled during summer when discharge is high. Conversely, storages are depleted during low flow in winter (Belz et al., 2007; Meile et al., 2011; Wesemann et al., 2018). A rough estimation supports this notion: Assuming the $1.86 \cdot 10^9$ m³ of storage being emptied between December and April (and filled between June and October), mean runoff would increase/decrease by approximately 10 m³/s/dec in these months during the investigated time frame, which corresponds to the trend magnitudes

depicted in Fig.4 b2. In addition to reservoir constructions, regulations of lakes levels and routing of rivers through lakes, e.g. the diversion of the Aare river into Lake Biel in 1887 (part of the First Jura-Waters Corrections), need to be considered (Wetter et al., 2011).

At gauge Wasserburg, pronounced changes in runoff seasonality do not show up until the second half of the 20th century (Fig.4 a3 and a4). In the Inn basin, the constructions of key reservoirs, such as the Gepatsch reservoir (Tyrol, Austria), the reser-

voir Lago di Livigno (Grisons, Switzerland and Lombardy, Italy) and the compensation reservoir Lai da Ova Spin (Grisons, Switzerland), were not completed until the 1960s. The construction of those big reservoirs coincides with the detected onset of changes in river runoff. We suspect that also in the Inn river basin, the construction and management of reservoirs for hydropower might be an important factor changing seasonality of river runoff. In addition to changes in seasonality, the operation of high-head hydropower stations causes unnatural fluctuation on (sub-)daily time scales (hydropeaking) (Meile et al., 2011;

Pérez Ciria et al., 2019) (see also Fig.F1). Effects of reservoirs possibly overlap (with) changes induced by changes in snow cover.

Also rainfall-runoff dominated rivers, such as the Main river at Wuerzburg, are strongly affected by hydro-engineering installations. One large-scale project inaugurated in 1992 after numerous decades of constructions represents the Rhine-Main-Danube waterway. In order to raise low water discharge in the Main river, about $1.55 \cdot 10^8$ m³ ($3,50 \cdot 10^8$ m³) of water are

transferred on average per year (in a dry year) from the Danube into the Main river basin via the Main-Danube-Canal (Maniak, 2016). The connection from the Rhine river until Wuerzburg with constructions of weirs to regulate the river´s water level was completed in the 1940s (Wirth, 1995). This onset of water level regulations in the 1940s coincides with increasing low QYEAs ($< 0.6$) at gauge Wuerzburg (Fig.4 d4). We suspected that anthropogenic alterations strongly impact the discharge of the Main river, particularly during low discharge periods. However, they seem to be insufficient to explain changes in higher QYEAs.

## 5.4   Changes in quantiles

We detect increasing high QYEAs at gauges Basel, Cologne and Wuerzburg (Fig.4 b4, c4 and d4). Possible driving mechanisms might be changes in precipitation: Our results hint at more (intense) rainfall in recent decades (Fig.5). In the following, we discuss possible underlying forcing mechanisms of detected signals. These include changes in large-scale circulation patterns, solar dimming/brightening and temperature-moisture feedbacks.

Long-term changes in the occurrence frequencies and/or characteristics of circulation pattern are known to have a strong impact on local climate. More frequent zonal circulation in winter since the 1970s, for example, might be responsible for "more frequent mild and humid winters in Central Europe" (Bárdossy and Caspary, 1990). This increase in zonal circulation follows upon several decades with increased numbers of blocking days during winter (Häkkinen et al., 2011). Blocking in the Atlantic region is anti-correlated with phases of the North Atlantic Oscillation (NAO) (Scherrer et al., 2006; Stein, 2000; Pavan





et al., 2000). Negative values of the NAO index "indicate periods of reduced north-south pressure gradient, reduced westerly winds and weaker advection of warm oceanic air onto the cold European landmass" (Parker et al., 2007). Wintertime NAO on the other hand is influenced by the Atlantic Multidecadal Oscillation (AMO): A "positive phase of the AMO results in more frequent negative NAO" (Peings and Magnusdottir, 2014). The AMO depicts multidecadal (60-70 years) variations in

sea surface temperatures in the North Atlantic basin (Peings and Magnusdottir, 2014; Kerr, 2000). After several decades of warm anomalies of sea surface temperatures (positive phase of AMO, which coincided with more frequent blocking days), the North Atlantic started to cool down and to transition into a negative AMO phase in the 1960s (Peings and Magnusdottir, 2014; Häkkinen et al., 2011). This transition into a negative AMO phase and less blocking days coincide with more frequent zonal circulation in winter (Bárdossy and Caspary, 1990), more (intense) rainfall (Fig.5 a3, b3, c3, a4, b4, c4) and an increase

in discharge, whereas changes seem to overlap with changes induced by anthropogenic alteration of the river network and changes in snow cover (Fig.4 b4, c4, d4).

Generally, detected pattern in temperature time series investigated in the framework of this study are consistent across all stations (Fig.5 and Fig. D1). In the case of precipitation, overall pattern are similar, however, stronger variations among stations show up. Precipitation is subject to stronger local and regional variability than temperature. This evidently limits the

informative value of precipitation recorded at individual points for discussions on catchment scale. However, variations of local meteorological variables are strongly influenced by large-scale flow and regional scale weather pattern (Scherrer et al., 2016; Murawski et al., 2018; Weusthoff, 2011). Weather pattern/types represent specific synoptic conditions and lead to certain meteorological condition in a region. For western Germany, precipitation correlates well on scales of hundreds of kilometers, particularly during winter (Schönwiese and Rapp, 1997). During summer, local convective storms are an important source of

rainfall and strong differences in amount and intensity over very short distances are possible (Sodemann and Zubler, 2010; Lavers et al., 2013). However, such convective storms 'are hardly of any relevance for the formation of floods in the large river basins of Central Europe, because the extent of convective rainstorms is restricted to local occurence' (Bronstert et al., 2007). Rainfall-runoff processes on larger scales are dominated by advective precipitation. The main moisture source then is the Atlantic ocean (Sodemann and Zubler, 2010). Following the above mentioned aspects, we hypothesize that even if

superimposed by local variability due to smaller-scale processes and regional variations due to general precipitation gradients, long-term signals in precipitation detected on point scale can provide important information for discussions on catchment scale. However, caution has to be exercised, results of available stations compared and findings not transferred to places outside the region of influence.

Marty (2008) relate detected shifts in snow days in Switzerland to an enhanced temperature increase due to changes in

circulation patterns coinciding with "the full magnitude of the greenhouse effect, which is no longer masked by solar dimming". Ater a multidecadal decrease from about the 1950s to the 1980s (solar dimming), recent decades saw an increase in regional solar irradiance (solar brightening) due to decreasing amounts of anthropogenic aerosols in the atmosphere (Ruckstuhl et al., 2008; Norris and Wild, 2007; Ruckstuhl and Norris, 2009).

Rapidly rising temperatures have the capacity to affect the entire hydrological cycle. A feedback mechanism being of major

importance in this respect is the temperature-moisture feedback: rising temperatures result in increasing evaporation and pre-





cipitation, which in turn leads to an intensification of the entire hydrological cycle (Huntington, 2006; Held and Soden, 2000). Lehmann et al. (2015) indicate that a "thermally driven moisture increase has significantly contributed to the intensification of extreme rainfalls since the 1980s".

Against the background of recent changes in temperature, precipitation, and frequencies in zonal circulation and following
Labat et al. (2004), we suggest that rapid increases in temperatures in recent decades result in an increased sea-land-transport of moisture and increases precipitation and runoff. Signals possibly overlap with changes in moisture transport due to varying frequencies in zonal circulation and blocking days.

At gauge Wasserburg, in contrast to other gauges investigated, high QYEAs do not increase. We suspect that in this case, also high discharges are controlled by snowmelt processes rather than liquid rainall. Furthermore, investigated rain gauges
might not depict changes in precipitation in the complex alpine topography of the catchment. In the High Rhine basin up to gauge Basel, large fractions of the basin are located outside the alpine ridge and liquid precipitation plays an important role throughout the year. Therefore, impacts of changes in snow cover, reservoir constructions and effects of more (intense) rainfall all seem to be detectable in measured discharge (Fig.4 b4). Changes in snow cover and river regulations, which decrease runoff seasonality, seem to primarily affect QYEAs below a level of 0.85 and more intense rainfall events increase the magnitude of
higher QYEAs.

## 6  Conclusions

We investigate daily observational data from key river gauges and meteorological stations located in Central Europe covering the time frame 1869-2016. Investigated time series stand out by the exceptional length and quality of their continuous recordings. A cascading sequence of analytical tools is used to extract high-resolution signals of long-term changes. In order
to acquire a comprehensive picture of long-term changes, we combine quantile sampling with moving average trend statistics and empirical mode decomposition. Given that the recordings have sufficient length and quality, presented tools enable investigations of high resolution and provide detailed insights into underlying trend patterns. A very high quality of the time series is required to prevent non-climatic factors, such as changes in observation practices or site relocation, to affect the determination of trends (Begert et al., 2005; Scherrer et al., 2016; Begert and Frei, 2018). A sufficient length of the time series is vital to be
able to distinguish between natural climate variability and signals of climate change.

The seasonality of the analyzed snow-dominated rivers decreases. We suspect river regulations, particularly reservoir constructions, to be the main driver of detected changes. Reservoirs are filled during summer when discharge is high and storages depleted during low flow in winter (e.g. Belz et al., 2007; Meile et al., 2011). In recent decades, runoff changes induced by reservoir constructions seem to overlap with, mostly anthropogenic, changes in snow cover (Laternser and Schneebeli, 2003;
Scherrer et al., 2004). Rising temperature reduce seasonal snow covers and the seasonal redistribution of runoff from winter to summer. An exact separation of effects of reservoirs and changes in snow cover and investigations of possible counterbalancing interactions are still pending and focus of future research. Furthermore, we suspect that detected decreases in discharge during

summer and autumn in the Lower Rhine region at gauge Cologne are the result of a downstream propagation of the alpine signal, possibly further overlapping with increasing evaporation rates in the basin.

In addition, our results hint at more (intense) rainfall in recent decades, particularly during winter. Detected changes in precipitation seem to intensify high discharges. The detected increase in precipitation (intensity) is not a gradual one over the entire time frame, but rather follows a U-shape (decline until the 1940s, then increase). Further research is necessary to pin down underlying mechanisms of detected changes in precipitation and runoff. We suspect that detected signals might be due to an increase in sea-land-transport of moisture, particularly during winter, being part of a recent intensification of the entire hydrological cycle (Huntington, 2006; Held and Soden, 2000; Lehmann et al., 2015; Labat et al., 2004). Temperature-driven increases in moisture and precipitation possibly overlap with natural multidecadal variations in sea-land-moisture transport (Parker et al., 2007; Kerr, 2000; Häkkinen et al., 2011; Peings and Magnusdottir, 2014; Bárdossy and Caspary, 1990; Scherrer et al., 2006; Pavan et al., 2000).

Over recent decades, hydrological regimes have been changing at a very fast pace. Some progress has been made in extracting long-terms signals of change in hydro-climatic data. However, further studies investigating long-term changes in river runoff focusing on the detection of underlying mechanisms and the disentanglement of their effects are of great urgency and importance.

*Data availability.* Climatological data used in this study was obtained from the Federal Office of Meteorology and Climatology of Switzerland, CH-8058 Zurich-Airport (MeteoSwiss). Discharge data analyzed was provided by The Global Runoff Data Centre, 56068 Koblenz, Germany (GRDC). Data analysis was carried out using the statistical software R (https://www.r-project.org/) (R Core Team, 2018).

*Author contributions.* ER conducted the analysis and wrote the manuscript. TF, GB and AB provided guidance in the process of data analysis and preparation of the manuscript.

*Competing interests.* The authors declare that they have no conflict of interest.

*Acknowledgements.* This research was funded by Deutsche Forschungsgemeinschaft within the research training group NatRiskChange (GRK 2043/1) at the University of Potsdam (https://www.uni-potsdam.de/natriskchange/). We thank the Federal Office of Meteorology and Climatology of Switzerland (MeteoSwiss), The Global Runoff Data Centre (GRDC) for providing climatological and discharge data, respectively. Furthermore, we thank the Copernicus Land Monitoring Service, implemented by the European Environmental Agency, for providing the European Digital Elevation Model (EU-DEM), version 1.1.





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



**Table 1.** Database studied: station name, associated river, location (WSG 84), altitude [m], daily resolution time series investigated with temperature (T), precipitation (P) and discharge (D), catchment area, mean runoff (MQ) and data source with Global Runoff Data Centre (GRDC) and Federal Office of Meteorology and Climatology of Switzerland (MeteoSwiss).

| Station | River | Lat. | Lon. | Alt. | Vari. | Area [km²] | MQ [m³/s] | Data source |
|---------|-------|------|------|------|-------|-----------|-----------|-------------|
| Basel Binningen | - | 47.5411 | 7.5836 | 316 | T-P | - | - | MeteoSwiss |
| Bern Zollikofen | - | 46.9908 | 7.4639 | 552 | T-P | - | - | MeteoSwiss |
| Zuerich Fluntern | - | 47.3781 | 8.5658 | 555 | T-P | - | - | MeteoSwiss |
| Wasserburg | Inn | 48.0593 | 12.2342 | 420 | D | $1.20 \cdot 10^4$ | 360 | GRDC |
| Basel Rheinhalle | Rhine | 47.5594 | 7.6167 | 294 | D | $3.59 \cdot 10^4$ | 1046 | GRDC |
| Cologne | Rhine | 50.9370 | 6.9633 | 35 | D | $1.44 \cdot 10^5$ | 2090 | GRDC |
| Wuerzburg | Main | 49.796 | 9.926 | 165 | D | $1.40 \cdot 10^4$ | 116 | GRDC |



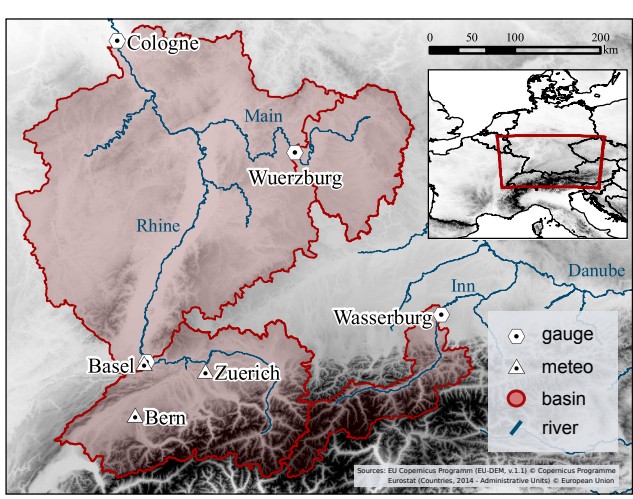

**Figure 1.** Topographic map of study area with location of river gauges, river basins and meteorological stations.

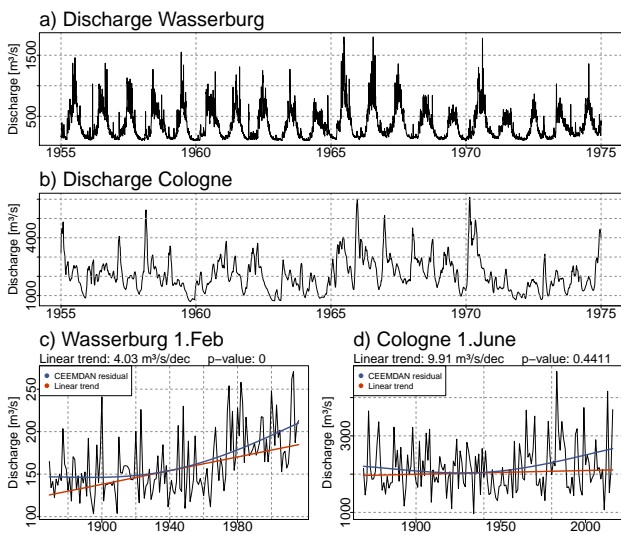

**Figure 2.** Discharge recordings from gauges Wasserburg (a) and Cologne (b) and measurements from all 1.February / 1.June for gauge Wasserburg / Cologne after applying a 30-day moving average filter over the entire time series (c/d). The robust Theil-Sen trend estimator and the Mann-Kendall trend test were applied to assess magnitude and significance of linear trends (red line). The CEEMDAN residual is used to extract the non-linear evolution of the trend (blue curve).

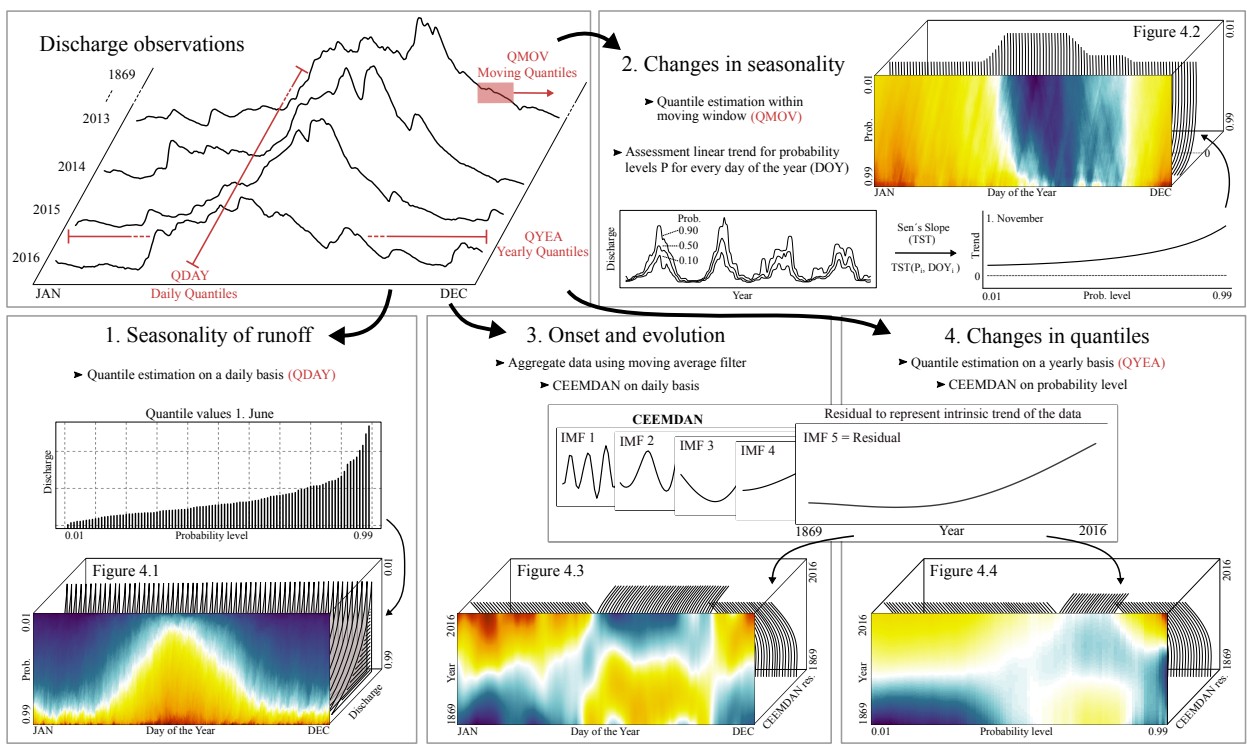

**Figure 3.** Schematic overview of analytical tools used to detect long-term changes in hydro-climatological time series. The analysis of discharge data is subdivided into four steps, where each step complements and extends the in the previous step acquired information.



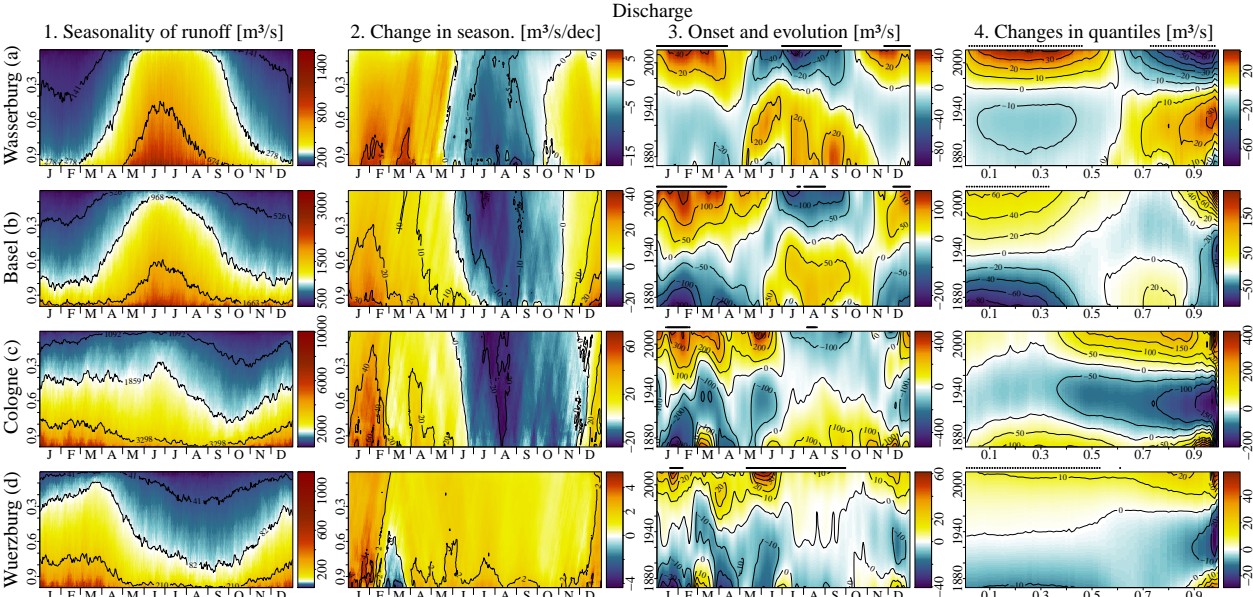

**Figure 4.** Seasonality of river runoff, change in seasonality, onsent and evolution of changes and changes in quantiles for discharge measured at gauges Wasserburg (a), Basel (b), Cologne (c) and Wuerzburg (d). Isolines in left panels '1. Seasonality of runoff' indicate quantiles for probabilities 0.1, 0.5 and 0.9 determined over the entire time series using all available measurements. Points on top of panels indicated days/probabilities with significant changes according to the Mann-Kendall trend test. Time frame investigated: 1869-2016.





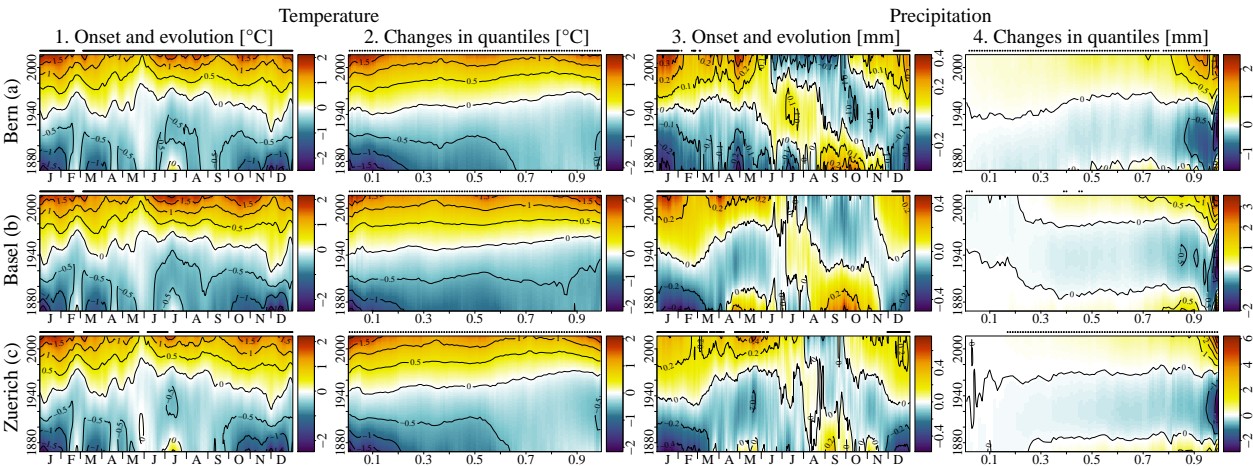

**Figure 5.** Onset and evolution of changes and changes in quantiles for temperature and precipitation measured at stations Bern (a), Basel (b) and Zuerich (c). Points on top of panels indicated days/probabilities with significant changes according to the Mann-Kendall trend test. Time frame investigated: 1869-2016.



**Table 2.** Summary of analysis results presented in Fig.4. Table arrangement reflects figure layout.

|  | 1. Seasonality of runoff | 2. Changes in seasonality | 3. Onset and evolution | 4. Changes in quantiles |
|---|---|---|---|---|
| a) Wasserburg | Snow dominated<br>Nival flow regime | Decrease seasonality<br>↑ Winter  ↓ Summer | 2nd half 20th century | ↑ Low  ↓ High |
| b) Basel | Snow dominated<br>Nival flow regime | Decrease seasonality<br>↑ Winter  ↓ Summer | Gradual change<br>Entire time frame | ↑ Low  ↓ High<br>↑ Very high |
| c) Cologne | Complex flow<br>Pluvio-nival | ↑ Winter and spring<br>↓ Summer and autumn | No clear onset<br>Nival + pluvial pattern | ↑ All prob. levels<br>U-shape |
| d) Wuerzburg | Rain-fed<br>Pluvial flow regime | ↑ All seasons | No clear onset | ↑ All prob. levels<br>Gradual + U-shape |





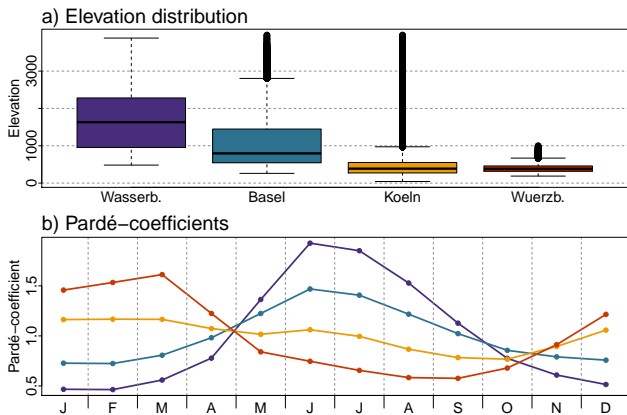

**Figure A1.** Elevation distribution (raster cells in 500 m resolution calculated based on EU-DEM v.1.1 by the EU Copernicus Programme) and Pardé-coefficients (mean monthly discharge divided by the mean annual discharge) (Pardé, 1933; Spreafico and Weingartner, 2005) for investigated river basins Wasserburg, Basel, Koeln and Wuerzburg.

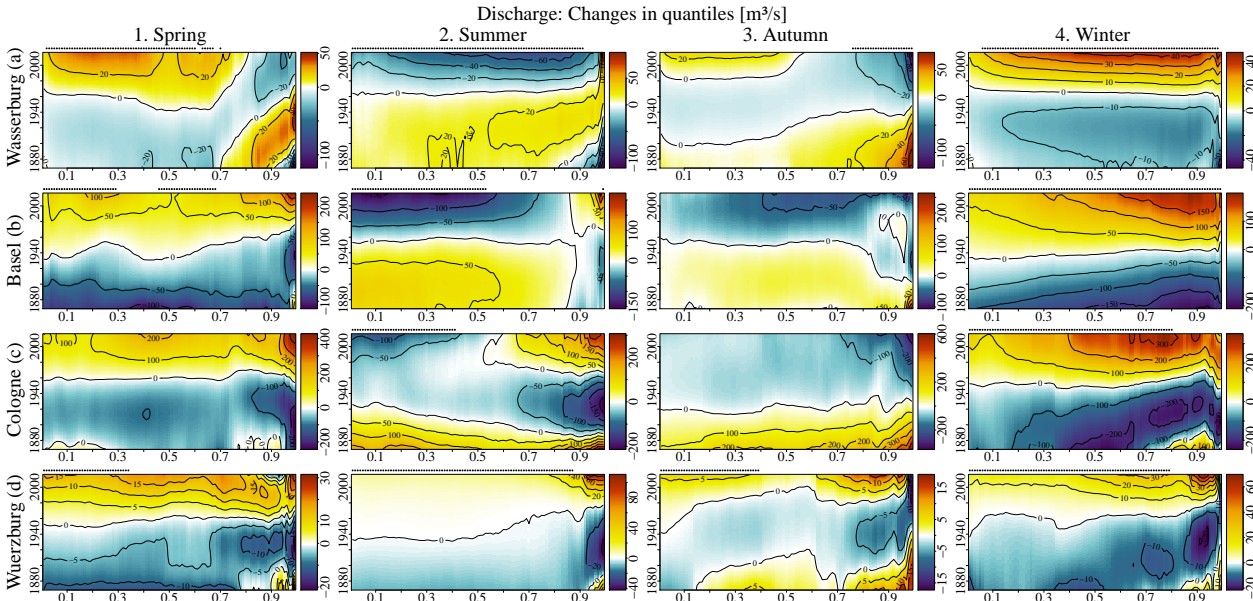

**Figure B1.** Changes in quantiles for individual seasons (Spring: March-May, Summer: June-August, Autumn: September-November and Winter: December-February) for discharge measured at gauges Wasserburg (a), Basel (b), Cologne (c) and Wuerzburg (d). Points on top of panels indicated days/probabilities with significant changes according to the Mann-Kendall trend test. Time frame investigated: 1869-2016.





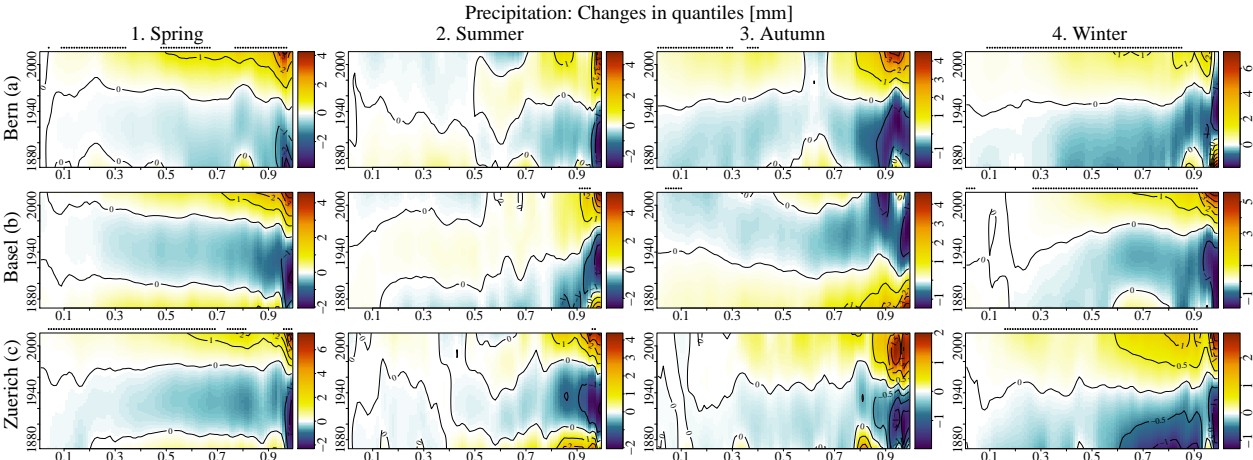

**Figure C1.** Changes in quantiles for individual seasons (Spring: March-May, Summer: June-August, Autumn: September-November and Winter: December-February) for precipitation measured at stations Bern (a), Basel (b) and Zuerich (c). Points on top of panels indicated days/probabilities with significant changes according to the Mann-Kendall trend test. Time frame investigated: 1869-2016.



**Figure D1.** Onset and evolution of changes and changes in quantiles for temperature and precipitation measured at stations Sion (a), Samedan (b), Neuchatel (c), Lugano (d), Geneve (e) and Chamont (f). Points on top of panels indicated days/probabilities with significant changes according to the Mann-Kendall trend test. Time frame investigated: 1869-2016.



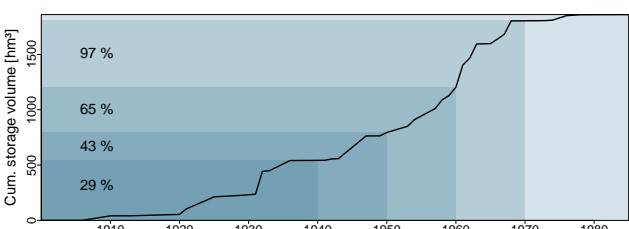

**Figure E1.** Cumulative storage volume of large storage lakes (active storage volume more than 0.3 hm³) in the High Rhine basin until gauge Basel. Figure is based on information presented in Wildenhahn and Klaholz (1996). Time frame displayed: 1900-1985.

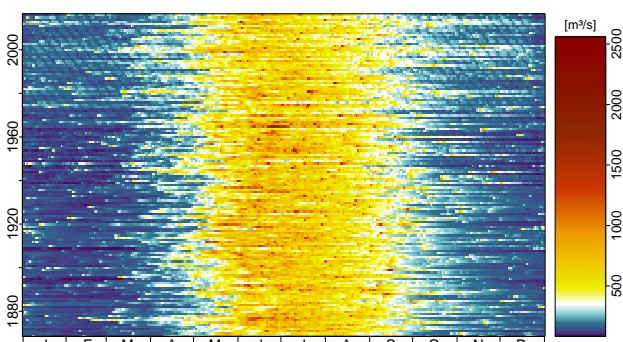

**Figure F1.** Raster hydrograph for gauge Wasserburg. In recent decades, hydropeaking (weekly pattern) due to the operation of high-head storage hydropower stations imprinted. Time frame displayed: 1869-2016.





**Table A1.** Additional climate stations investigated: station name, location (WSG 84), altitude [m], daily resolution time series investigated with temperature (T) and precipitation (P) and data source Federal Office of Meteorology and Climatology of Switzerland (MeteoSwiss).

| Station | Lat. | Lon. | Alt. | Vari. | Data source |
|---|---|---|---|---|---|
| Sion | 46.2186 | 7.3303 | 482 | T-P | MeteoSwiss |
| Samedan | 46.5264 | 9.8789 | 1709 | T-P | MeteoSwiss |
| Neuchatel | 47.0000 | 6.9533 | 485 | T-P | MeteoSwiss |
| Lugano | 46.0042 | 8.9602 | 273 | T-P | MeteoSwiss |
| Geneve / Cointrin | 46.2475 | 6.1278 | 411 | T-P | MeteoSwiss |
| Chaumont | 47.0492 | 6.9789 | 1136 | T-P | MeteoSwiss |