# Peer review of "Long-term changes in Central European river discharge 1869-2016: impact of changing snow covers, reservoir constructions and an intensified hydrological cycle"

_Hydrology and Earth System Sciences, 2019_

## Referee Comment (RC1) · Anonymous Referee #1 · 1 Dec 2019

The manuscript analyses river discharge and climate data from a number of different tributaries and gauges along the Rhine river in order to describe and analyse possible changes in runoff regimes and their controlling factors/processes. Of particular interest is the focus on stations where long term data sets are available.

General Comment

In general, I find the manuscript well organised, and especially the strict separation of result description and discussion I like. It is easy to read, and as far as I am able

to judge as a non-native speaker, well written. The topic of the manuscript is highly relevant and will provide and important contribution to the current discourse and climate change impact on hydrological system, that is worth being published in HESS.

Specific Comments, that needs urgently to be addressed:

- The introduction is very compact (nothing wrong with that), but should explore and elaborate a bit more on the weaknesses of current approaches to analyse long time series of data. The statement (page2, line 22) ". . . new sets of analytical tools to extract information stored in this times series needs to be developed, tested, and applied . . ." is not justified by a critical review of currently available methods. Also, the advantage of then developed (own developments??) or applied methods to existing ones needs to be demonstrated.

- You use many abreviations – as far as I can see they are all properly introduced/defined the first time mentioned, but a list of symbols/abbreviations would very much help, especially when reading the manunscript over longer time periods.

- The calculation of QMOV is not fully clear to me. From Fig.3 and section 3.1 (Discharge observations) it looks like daily quantiles are calculated from 148 values (single date, 1869-2016). How are "Changes in Seasonality" calculated – section 3.2 mentions the application of a 30day moving window. Is it operation on the previously extracted daily quantiles or does it operate on the daily runoff values and the quantiles are calculated from there. In my opion there would be arguments for both ways. You should also clearify than when you filter the data are highly correlated and I am not sure whether the TST estimator is made for these conditions. Please clearify and add some information on this.

Minor Comments:

P2, l28-30: Why focus on snow cover, as a hydrologist I would be mor interested on the snow water equivalent.

P3, l13: It would be good to hear something about the test for homogeneity.

P3. L5ff: Please use dot for separating decimal figures throughout the manuscript (1.20 10ˆ4)

P5, l2: which plots are addressed here.

P5, l13: How you define flood?

P11, l29: "anthropogenic" - is this a statement that CC is mainly driven by men, otherwise I would CC-driven changes.

In my opinion, the paper needs some moderate revisions as mentioned in the specific comments which would significantly improve the paper and make it suitable for publication in HESS.

---

## Referee Comment (RC2) · Massimiliano Zappa (Referee) · 5 Dec 2019

[revised manuscript text omitted]

Nival flow regime | Decrease seasonality
↑ Winter ↓ Summer | Gradual change
Entire time frame | ↑ Low ↓ High
↑ Very high |
| c) Cologne | Complex flow
Pluvio-nival | ↑ Winter and spring
↓ Summer and autumn | No clear onset
Nival + pluvial pattern | ↑ All prob. levels
U-shape |
| d) Wuerzburg | Rain-fed
Pluvial flow regime | ↑ All seasons | No clear onset | ↑ All prob. levels
Gradual + U-shape |

[Figure]

[Figure]

[Figure]

**Figure A1.** Elevation distribution (raster cells in 500 m resolution calculated based on EU-DEM v.1.1 by the EU Copernicus Programme) and Pardé-coefficients (mean monthly discharge divided by the mean annual discharge) (Pardé, 1933; Spreafico and Weingartner, 2005) for investigated river basins Wasserburg, Basel, Koeln and Wuerzburg.

[Figure]

**Figure B1.** Changes in quantiles for individual seasons (Spring: March-May, Summer: June-August, Autumn: September-November and Winter: December-February) for discharge measured at gauges Wasserburg (a), Basel (b), Cologne (c) and Wuerzburg (d). Points on top of panels indicated days/probabilities with significant changes according to the Mann-Kendall trend test. Time frame investigated: 1869-2016.

[Figure]

[Figure]

[Figure]

**Figure C1.** Changes in quantiles for individual seasons (Spring: March-May, Summer: June-August, Autumn: September-November and Winter: December-February) for precipitation measured at stations Bern (a), Basel (b) and Zuerich (c). Points on top of panels indicated days/probabilities with significant changes according to the Mann-Kendall trend test. Time frame investigated: 1869-2016.

**Figure D1.** Onset and evolution of changes and changes in quantiles for temperature and precipitation measured at stations Sion (a), Samedan (b), Neuchatel (c), Lugano (d), Geneve (e) and Chamont (f). Points on top of panels indicated days/probabilities with significant changes according to the Mann-Kendall trend test. Time frame investigated: 1869-2016.

[Figure]

[Figure]

[Figure]

**Figure E1.** Cumulative storage volume of large storage lakes (active storage volume more than 0.3 hm³) in the High Rhine basin until gauge Basel. Figure is based on information presented in Wildenhahn and Klaholz (1996). Time frame displayed: 1900-1985.

[Figure]

**Figure F1.** Raster hydrograph for gauge Wasserburg. In recent decades, hydropeaking (weekly pattern) due to the operation of high-head storage hydropower stations imprinted. Time frame displayed: 1869-2016.

[Figure]

[Figure]

**Table A1.** Additional climate stations investigated: station name, location (WSG 84), altitude [m], daily resolution time series investigated with temperature (T) and precipitation (P) and data source Federal Office of Meteorology and Climatology of Switzerland (MeteoSwiss).

| Station | Lat. | Lon. | Alt. | Vari. | Data source |
|---|---|---|---|---|---|
| Sion | 46.2186 | 7.3303 | 482 | T-P | MeteoSwiss |
| Samedan | 46.5264 | 9.8789 | 1709 | T-P | MeteoSwiss |
| Neuchatel | 47.0000 | 6.9533 | 485 | T-P | MeteoSwiss |
| Lugano | 46.0042 | 8.9602 | 273 | T-P | MeteoSwiss |
| Geneve / Cointrin | 46.2475 | 6.1278 | 411 | T-P | MeteoSwiss |
| Chaumont | 47.0492 | 6.9789 | 1136 | T-P | MeteoSwiss |

---

## Author Comment (AC1) · 19 Dec 2019

**hess-2019-487**
**Response to anonymous referee #1**

**Erwin Rottler, Till Francke, Gerd Bürger and Axel Bronstert**

December 19, 2019

Dear Anonymous Referee,

thank you very much for reviewing our manuscript. We are very grateful for your comments and suggestions. In the following, detailed responses to all your comments.

On behalf of all authors,

sincerly,

Erwin Rottler

**Contents**

**1  Specific comments**

**1.1  Comment 1**

> The introduction is very compact (nothing wrong with that), but should explore
> and elaborate a bit more on the weaknesses of current approaches to analyse
> long time series of data. The statement (page2, line 22) ". . . new sets of
> analytical tools to extract information stored in this times series needs to be
> developed, tested, and applied . . ." is not justified by a critical review of currently
> available methods. Also, the advantage of then developed (own developments??)
> or applied methods to existing ones needs to be demonstrated.

Indeed, an overview of traditionally used methods to analyse long time series and their weak-
nesses/strengths is missing. We tried to point at the advantages and limitations of our ap-
proach, but you are right, we need to put this into context and formulate it more precise. We
will prepare respective paragraphs and incorporate it into the introduction and discussion.

**1.2  Comment 2**

> You use many abreviations – as far as I can see they are all properly
> introduced/defined the first time mentioned, but a list of symbols/abbreviations
> would very much help, especially when reading the manunscript over longer
> time periods.

We will work through our manuscript to see what abbreviations are actually needed. Yes, a list
could be useful, so the reader can look them up easily if needed. Thank you for mentioning
this.

**1.3  Comment 3**

> The calculation of QMOV is not fully clear to me. From Fig.3 and section 3.1
> (Discharge observations) it looks like daily quantiles are calculated from 148
> values (single date, 1869-2016). How are "Changes in Seasonality" calculated –
> section 3.2 mentions the application of a 30day moving window. Is it operation
> on the previously extracted daily quantiles or does it operate on the daily runoff
> values and the quantiles are calculated from there. In my opion there would be
> arguments for both ways. You should also clearify than when you filter the data
> are highly correlated and I am not sure whether the TST estimator is made for
> these conditions. Please clearify and add some information on this.

Quantile estimations on a daily basis (QDAY) and quantile estimation within a moving win-
dow (QMOV) are two independent steps. QMOV operates on discharge data. In a way, QMOV

is similar to QYEA (quantile estimations using all values of one year), but with QMOV we only use part of the values of a year, the one within a 30-day window. We realize that we need to improve our description in the method section to make sure our approach is understood more easily. We will clarify and add some more information. Thank you very much for the hint.

**2 Minor comments**

**P2, l28-30: Why focus on snow cover, as a hydrologist I would be mor interested on the snow water equivalent.**

With 'snow cover' we do not think of the areal extent of snow, but indeed changes in water being stored in the temporary snow cover. It seems the wording we chose is not precise enough. We will think of a formulation that describes our ideas more clear and prevents any ambiguity.

**P3, l13: It would be good to hear something about the test for homogeneity.**

We will try to find out more about the tests conducted by [Pfister et al., 2006] that made them state that the time series can be considered homogeneous.

**P3. L5ff: Please use dot for separating decimal figures throughout the manuscript (1.20 10^4)**

Thank you for pointing this out. We will replace commas with dots.

**P5, l2: which plots are addressed here.**

We address the two right columns of Fig. 4, Yes, you are right, we need to mention this here specifically and refer to the figure.

**P5, l13: How you define flood?**

We do not specifically define 'flood' in our paper, yet. Yes, you are right, we need to better explain what our quantile estimates represent and, more importantly, what they do not. With regard to floods, it might help to discuss quantile values in relation to return periods and extreme value statistics. We will work on this and include it in our manuscript.

**P11, l29: "anthropogenic" - is this a statement that CC is mainly driven by men, otherwise I would CC-driven changes.**

We included the 'anthropogenic' here to emphasise that recent changes in snow cover are not due to large-scale climate variability (which are important to understand changes in alpine snow cover, see e.g. [Scherrer et al., 2004]), but due to rising temperatures being part of recent climatic changes. We will reformulate corresponding sentences to make them more clear.

**References**

[Pfister et al., 2006] Pfister, C., Weingartner, R., and Luterbacher, J. (2006). Hydrological winter droughts over the last 450 years in the Upper Rhine basin: A methodological approach. *Hydrological Sciences Journal*, 51(5):966–985.

[Scherrer et al., 2004] Scherrer, S. C., Appenzeller, C., and Laternser, M. (2004). Trends in Swiss Alpine snow days: The role of local- and large-scale climate variability. *Geophysical Research Letters*, 31(13).

---

## Author Comment (AC2) · 19 Dec 2019

**hess-2019-487**
**Response to Massimiliano Zappa (referee)**

**Erwin Rottler, Till Francke, Gerd Bürger and Axel Bronstert**

December 19, 2019

Dear Mr Zappa,

thank you very much for reviewing our manuscript. We are very grateful for your comments and suggestions. In the following, detailed responses to all your comments.

On behalf of all authors,

sincerly,

Erwin Rottler

**Contents**

**1 Comments on Abstract and Introduction**

**1.1 Page 1, Line 9-11**

**No real surpize, well supported by data. You might find interest in this paper to support this finding [Farinotti et al., 2016]**

I very much enjoyed reading [Farinotti et al., 2016]. It provides interesting information that will help us to put our findings into context.

**1.2 Page 2, Line 1**

**Here some classifications on changing snowmelt are presented in a climate impact framework [Speich et al., 2015]**

Looks interesting. We are happy to include this information into our manuscript.

**1.3 Page 2, Line 25**

**I miss some recent papers here. They focus on flood, but might offer information for discussion in your study. [Berghuijs et al., 2019, Blöschl et al., 2017, Blöschl et al., 2019]**

This information will help us to improve our discussion.

**1.4 Page 3, Line 2**

**Relatively small data basis**

In addition to the three meteorological stations we present in the main manuscript, we include results of further meteorological stations into the appendix. Unfortunately, there are not many more recordings covering such a long time frame and having such high quality at the same time. With regard to discharge, we initially looked at other gauges as well and in an earlier version e.g. also included gauges Dresden (Elbe river) and Burghausen (Salzach river). However, this turned out to shift the focus away from what we wanted to discuss and made it very difficult to prepare a concise manuscript. In our study, we focus on the analysis of long and consistent time series. Therefore, it seems that, to a certain extent, we have to accept trade-offs in the number of stations included.

**2 Comments on Methods**

**2.1 Page 3, Line 31-33**

**Very nice and useful graphical abstract**

Thank you! It took us some time to come up with a proper graphical illustration to support our analysis tools.

**2.2 Page 4, Line 2**

**Just a clarification here. You make the quantiles in a shape manner for every DOY and not for a window centered on every DOY. We have good experience with quantiles centered on +/- 15 Days for evry DOY. It gives more smooth regimes for pluvial basins. For large basins as yours this might not be necessary.**

In step 1 of our analysis no moving windows or other averaging techniques are applied. We very much would like to keep it that way and avoid calculating averages before quantile estimations. Yes, most probably the size of the basins and the length of the time series (!) help that no prior averaging is necessary.

**2.3 Page 4, Line 5-6**

**Thanks for this technical indication.**

You're welcome.

**2.4 Page 4, Line 8**

**Here you use the 30 days, but only to create a smoothed time series. As it is formulated, it can also be what I was describing in my comment above.**

We do not use the window to smooth the data, but calculate quantiles within the moving window. Yes, it was also possible to average using a moving window prior to the determination of quantiles on a daily basis (QDAY). We will try to improve the description of our approach, particularly in section 3.2, as we still need to do a better job there.

**2.5 Page 4, Line 29**

**Any sensitivity test prior to choosing these 30 and 90 days windows to report?**

We made good experiences with moving average trend statistics using a 30 day window (see e.g. [Kormann et al., 2015, Rottler et al., 2019]). Monthly values provide stability and still enable the calculation of trends in a highly resolved manner. For precipitation, which is more variable than temperature or discharge, we had to increase the window size to get necessary stability. We tried different window sizes varying them by hand in our scripts. We did not conduct specific sensitivity tests/analysis, but settled on commonly used and established monthly (30) and seasonal (90) values. Our testing indicated that those window sizes also perform very well in our analysis. Additional information we think of including into the manuscript on this issue could sound like: 'We conducted tests with varying window sizes. These tests indicated that 30 (90) day constitute a good compromise between robustness of the signal and preservation of the signal variability.' We will work on this to further improve our manuscript.

**3 Comments on Results and Discussion**

**3.1 Page 5, Line 9**

**Are you really plotting discharge with lowflow on the top of the y-axis and peak discharge on the bottom-part of the y-axis?!?!? This is absolutely contra intuitive! Change! Furthermore, I would plot the dry season in red and the wet one in blue .....**

Indeed. Thank you for pointing at this. Reversing the y-axis/colors makes the understanding of our figures way more intuitive. We tried already and it looks better now. We will change our figures accordingly.

**3.2 Page 5, Line 14**

**This has to do with size. For such large rivers is in my opinion quite challenging to attribute a distinct regime characteristic. I think that classic regime classifications (nival, glacial, pluvial) is something you can attribute to mesoscale basins (up to 1000 km2 or so).**

Yes, the larger basins are, the more difficult it usually gets. We will rethink our regime descriptions and try to formulate a more suitable characterization.

**3.3 Page 5, Line 23**

**I see it, very nice!**

Cool.

**3.4 Page 6, Line 1-6**

**Have you tought to createa proxy for liquid/solid precipitation and combining P and T? For the Rhine in Basel an additional station at elevation > 1000 m might be useful (Davos?)**

We did not think of creating a liquid/solid precipitation proxy yet. It indeed might provide very interesting supporting information for our discussion. We will look into this. We did not include station Davos into our study as it did not fulfil our criterion that time series should not have data gaps longer than 60 days. But we will look at the time series again and try. Maybe also stations Samedan (1709 m) or Chaumant (1136 m) turn out to be useful in this regard.

**3.5 Page 6, Line 19**

**Too few stations to make any speculation on that regard**

Yes, the limited number of stations available with such data length and quality limits the significance of our results. This might be the most vulnerable point of our analysis. However, consistent analysis results for stations and parameters investigated make us confident that

even if the number of stations is limited, we attain meaningful results that are worth discussing. We recognize that we have to be very careful on what speculations to make and to clearly indicated what to be a robust finding of our analysis and what just speculation. We will work through our manuscript again to improve our writing in this regard.

**3.6 Page 7, Line 25**

**Also [Marty et al., 2017]**

Thank you. We will include it into our manuscript.

**3.7 Page 8, Line 25**

**See, in German https://hydrologischeratlas.ch/produkte/druckausgabe/fliessgewasser-und-seen/tafel-5-3 Fig. 1 https://hydrologischeratlas.ch/downloads/01/content/Tafel_53.pdf**

Very nice maps! We were not aware of their existence. This information helps us to better understand the influence of hydropower and lakes on river runoff.

**3.8 Page 8, Line 31**

**See also [Bosshard et al., 2013]**

We will include information of the mentioned work into our manuscript.

**References**

[Berghuijs et al., 2019] Berghuijs, W. R., Harrigan, S., Molnar, P., Slater, L. J., and Kirchner, J. W. (2019). The Relative Importance of Different Flood-Generating Mechanisms Across Europe. *Water Resources Research*, 55(6):4582–4593.

[Blöschl et al., 2017] Blöschl, G., Hall, J., Parajka, J., Perdigão, R. A. P., Merz, B., Arheimer, B., Aronica, G. T., Bilibashi, A., Bonacci, O., Borga, M., Čanjevac, I., Castellarin, A., Chirico, G. B., Claps, P., Fiala, K., Frolova, N., Gorbachova, L., Gül, A., Hannaford, J., Harrigan, S., Kireeva, M., Kiss, A., Kjeldsen, T. R., Kohnová, S., Koskela, J. J., Ledvinka, O., Macdonald, N., Mavrova-Guirguinova, M., Mediero, L., Merz, R., Molnar, P., Montanari, A., Murphy, C., Osuch, M., Ovcharuk, V., Radevski, I., Rogger, M., Salinas, J. L., Sauquet, E., Šraj, M., Szolgay, J., Viglione, A., Volpi, E., Wilson, D., Zaimi, K., and Živković, N. (2017). Changing climate shifts timing of European floods. *Science*, 357(6351):588–590.

[Blöschl et al., 2019] Blöschl, G., Hall, J., Viglione, A., Perdigão, R. A. P., Parajka, J., Merz, B., Lun, D., Arheimer, B., Aronica, G. T., Bilibashi, A., Boháč, M., Bonacci, O., Borga, M., Čanjevac, I., Castellarin, A., Chirico, G. B., Claps, P., Frolova, N., Ganora, D., Gorbachova, L., Gül, A., Hannaford, J., Harrigan, S., Kireeva, M., Kiss, A., Kjeldsen, T. R., Kohnová, S., Koskela, J. J., Ledvinka, O., Macdonald, N., Mavrova-Guirguinova, M., Mediero, L., Merz, R., Molnar,

P., Montanari, A., Murphy, C., Osuch, M., Ovcharuk, V., Radevski, I., Salinas, J. L., Sauquet, E., Šraj, M., Szolgay, J., Volpi, E., Wilson, D., Zaimi, K., and Živković, N. (2019). Changing climate both increases and decreases European river floods. *Nature*, 573(7772):108–111.

[Bosshard et al., 2013] Bosshard, T., Carambia, M., Goergen, K., Kotlarski, S., Krahe, P, Zappa, M., and Schär, C. (2013). Quantifying uncertainty sources in an ensemble of hydrological climate-impact projections. *Water Resources Research*, 49(3):1523–1536.

[Farinotti et al., 2016] Farinotti, D., Pistocchi, A., and Huss, M. (2016). From dwindling ice to headwater lakes: could dams replace glaciers in the European Alps? *Environmental Research Letters*, 11(5):054022.

[Kormann et al., 2015] Kormann, C., Francke, T., Renner, M., and Bronstert, A. (2015). Attribution of high resolution streamflow trends in Western Austria: An approach based on climate and discharge station data. *Hydrology and Earth System Sciences*, 19(3):1225–1245.

[Marty et al., 2017] Marty, C., Schlögl, S., Bavay, M., and Lehning, M. (2017). How much can we save? Impact of different emission scenarios on future snow cover in the Alps. *The Cryosphere*, 11(1):517–529.

[Rottler et al., 2019] Rottler, E., Kormann, C., Francke, T., and Bronstert, A. (2019). Elevation-dependent warming in the Swiss Alps 1981–2017: Features, forcings and feedbacks. *International Journal of Climatology*, 39(5):2556–2568.

[Speich et al., 2015] Speich, M. J. R., Bernhard, L., Teuling, A. J., and Zappa, M. (2015). Application of bivariate mapping for hydrological classification and analysis of temporal change and scale effects in Switzerland. *Journal of Hydrology*, 523:804–821.

---

## Author Response (AR1)

**hess-2019-487**
**Revision II**

**Erwin Rottler, Till Francke, Gerd Bürger and Axel Bronstert**

**January 30, 2020**

Dear Mr Freer,

we are very happy to get the chance to revise our manuscript. In the following, we list all comments of the two reviewers, our responses and how we changed our manuscript accordingly. When specifying page/line numbers, we refer to the 'Marked-up manuscript version' below, where all additions and deletions in comparison to the previous version are marked. Should you have any further questions or comments, please do not hesitate to contact us. Thank you very much for your support!

On behalf of all authors,

sincerly,

Erwin Rottler

**Contents**

**1 Referee 1 (Anonymous)**

**1.1 Specific comments**

**1.1.1 Comment 1**

> The introduction is very compact (nothing wrong with that), but should explore and elaborate a bit more on the weaknesses of current approaches to analyse long time series of data. The statement (page2, line 22) ". . . new sets of analytical tools to extract information stored in this times series needs to be developed, tested, and applied . . ." is not justified by a critical review of currently available methods. Also, the advantage of then developed (own developments??) or applied methods to existing ones needs to be demonstrated.

Indeed, an overview of traditionally used methods to analyse long time series and their weaknesses/strengths is missing. We tried to point at the advantages and limitations of our approach, but you are right, we need to put this into context and formulate it more precise. We will prepare respective paragraphs and incorporate it into the introduction and discussion.

We extended the paragraph in the introduction that puts our analysis into the context of other studies. We tried to point at the need for further analytical tools by comparing with commonly used linear regression approach, e.g. the non-parametric Mann-Kendall trend test (Page 2, Line 20-27).

**1.1.2 Comment 2**

> You use many abreviations – as far as I can see they are all properly introduced/defined the first time mentioned, but a list of symbols/abbreviations would very much help, especially when reading the manunscript over longer time periods.

We will work through our manuscript to see what abbreviations are actually needed. Yes, a list could be useful, so the reader can look them up easily if needed. Thank you for mentioning this.

We added a list with all abbreviations and acronyms used to the appendix (Tab. A1) and mention its existence at the beginning of the method description, just before a lot of them are used (Page 4, Line 8-9).

**1.1.3 Comment 3**

> The calculation of QMOV is not fully clear to me. From Fig.3 and section 3.1 (Discharge observations) it looks like daily quantiles are calculated from 148 values (single date, 1869-2016). How are "Changes in Seasonality" calculated – section 3.2 mentions the application of a 30day moving window. Is it operation on the previously extracted daily quantiles or does it operate on the daily runoff values and the quantiles are calculated from there. In my opion there would be arguments for both ways. You should also clearify than when you filter the data are highly correlated and I am not sure whether the TST estimator is made for these conditions. Please clearify and add some information on this.

Quantile estimations on a daily basis (QDAY) and quantile estimation within a moving window (QMOV) are two independent steps. QMOV operates on discharge data. In a way, QMOV is similar to QYEA (quantile estimations using all values of one year), but with QMOV we only use part of the values of a year, the one within a 30-day window. We realize that we need to improve our description in the method section to make sure our approach is understood more easily. We will clarify and add some more information. Thank you very much for the hint.

We extended the description of QMOV (Page 4, Line 17-25). This hopefully helps the avoid misunderstandings.

**1.2 Minor comments**

**P2, l28-30: Why focus on snow cover, as a hydrologist I would be mor interested on the snow water equivalent.**

With 'snow cover' we do not think of the areal extent of snow, but indeed changes in water being stored in the temporary snow cover. It seems the wording we chose is not precise enough. We will think of a formulation that describes our ideas more clear and prevents any ambiguity.

We scanned through our manuscript and changed sentences that might cause misunderstandings. 'Snow pack' often seems to be the better choice (see e.g. Page 2, Line 6 or Page 7 Line 11).

**P3, l13: It would be good to hear something about the test for homogeneity.**

We will try to find out more about the tests conducted by [Pfister et al., 2006] that made them state that the time series can be considered homogeneous.

I the last paragraph (bottom third) on page 696, [Pfister et al., 2006] describe the history and quality of data recorded at gauge Basel in detail. Digital values are available since 1869 (the time series even dates back until 1808) and it is statistical tests on daily runoff means that make them conclude that the time series is homogeneous for the digital part (Page 3, Line 14-16).

**P3. L5ff: Please use dot for separating decimal figures throughout the manuscript (1.20 10^4)**

Thank you for pointing this out. We will replace commas with dots.

Done. See e.g. Page 3, Line 7, 12 and 13.

**P5, l2: which plots are addressed here.**

We address the two right columns of Fig. 4, Yes, you are right, we need to mention this here specifically and refer to the figure.

We added respective figure reference (Page 5, Line 17).

**P5, l13: How you define flood?**

We do not specifically define 'flood' in our paper, yet. Yes, you are right, we need to better explain what our quantile estimates represent and, more importantly, what they do not. With regard to floods, it might help to discuss quantile values in relation to return periods and extreme value statistics. We will work on this and include it in our manuscript.

We rephrased this paragraph (Page 5, Line 25-30). As our analytical tools do not provide direct information on floods, i.e. events with long return period, we try to rather talk about 'high runoff' when describing our results. We use the word 'flood' only when refering to other studies, which address them specifically.

**P11, l29: "anthropogenic" - is this a statement that CC is mainly driven by men, otherwise I would CC-driven changes.**

We included the 'anthropogenic' here to emphasise that recent changes in snow cover are not due to large-scale climate variability (which are important to understand changes in alpine snow cover, see e.g. [Scherrer et al., 2004]), but due to rising temperatures being part of recent climatic changes. We will reformulate corresponding sentences to make them more clear.

In order to avoid any misunderstandings, we changed the wording of the sentence (Page 12, Line 22).

**2 Referee 2 (Mr Zappa)**

**2.1 Comments on Abstract and Introduction**

**2.1.1 Page 1, Line 9-11**

**No real surpize, well supported by data. You might find interest in this paper to support this finding [Farinotti et al., 2016]**

I very much enjoyed reading [Farinotti et al., 2016]. It provides interesting information that will help us to put our findings into context.

Included Page 9, Line 23.

**2.1.2 Page 2, Line 1**

**Here some classifications on changing snowmelt are presented in a climate impact framework [Speich et al., 2015]**

Looks interesting. We are happy to include this information into our manuscript.

Included Page 7, Line 18.

**2.1.3 Page 2, Line 25**

**I miss some recent papers here. They focus on flood, but might offer information for discussion in your study. [Berghuijs et al., 2019, Blöschl et al., 2017, Blöschl et al., 2019]**

This information will help us to improve our discussion.

Included Page 7, Line 27-28.

**2.1.4 Page 3, Line 2**

**Relatively small data basis**

In addition to the three meteorological stations we present in the main manuscript, we include results of further meteorological stations into the appendix. Unfortunately, there are not many more recordings covering such a long time frame and having such high quality at the same time. With regard to discharge, we initially looked at other gauges as well and in an earlier version e.g. also included gauges Dresden (Elbe river) and Burghausen (Salzach river). However, this turned out to shift the focus away from what we wanted to discuss and made it very difficult to prepare a concise manuscript. In our study, we focus on the analysis of long and consistent time series. Therefore, it seems that, to a certain extent, we have to accept trade-offs in the number of stations included.

To call attention to the limited amount of stations available/presented, we extended our manuscript. We added further reasoning for our approach into the chapter 'Study area and Data' (Page 3, Line 30 - Page 4, Line 2) and extended the 'Conclusion' (Page 14, Line 14-18). We change 'detect a similar pattern' to 'similar pattern seem to show up' (e.g. Page 7, Line 3).

**2.2 Comments on Methods**

**2.2.1 Page 3, Line 31-33**

**Very nice and useful graphical abstract**

Thank you! It took us some time to come up with a proper graphical illustration to support our analysis tools.

We updated the graphical illustration with figures having the new reversed color scale (see Fig.3).

**2.2.2 Page 4, Line 2**

**Just a clarification here. You make the quantiles in a shape manner for every DOY and not for a window centered on every DOY. We have good experience with quantiles centered on +/- 15 Days for evry DOY. It gives more smooth regimes for pluvial basins. For large basins as yours this might not be necessary.**

In step 1 of our analysis no moving windows or other averaging techniques are applied. We very much would like to keep it that way and avoid calculating averages before quantile estimations. Yes, most probably the size of the basins and the length of the time series (!) help that no prior averaging is necessary.

We tried calculating QDAYs (Quantiles on a daily basis) after applying moving average filters. This step seems not to be necessary in our case. The time series are long enough that a clear picture of runoff seasonality can form, also for the rain-fed basins. But we will keep it in mind for future analysis.

**2.2.3 Page 4, Line 5-6**

**Thanks for this technical indication.**

You're welcome.

**2.2.4 Page 4, Line 8**

**Here you use the 30 days, but only to create a smoothed time series. As it is formulated, it can also be what I was describing in my comment above.**

We do not use the window to smooth the data, but calculate quantiles within the moving window. Yes, it was also possible to average using a moving window prior to the determination of quantiles on a daily basis (QDAY). We will try to improve the description of our approach, particularly in section 3.2, as we still need to do a better job there.

We included a few more lines of description into the method section (Page 4, Line 17-25) We also tried calculating QDAYs after calculating moving averages (see comment above).

**2.2.5 Page 4, Line 29**

**Any sensitivity test prior to choosing these 30 and 90 days windows to report?**

We made good experiences with moving average trend statistics using a 30 day window (see e.g. [Kormann et al., 2015, Rottler et al., 2019]). Monthly values provide stability and still enable the calculation of trends in a highly resolved manner. For precipitation, which is more variable than temperature or discharge, we had to increase the window size to get necessary

stability. We tried different window sizes varying them by hand in our scripts. We did not conduct specific sensitivity tests/analysis, but settled on commonly used and established monthly (30) and seasonal (90) values. Our testing indicated that those window sizes also perform very well in our analysis. Additional information we think of including into the manuscript on this issue could sound like: 'We conducted tests with varying window sizes. These tests indicated that 30 (90) day constitute a good compromise between robustness of the signal and preservation of the signal variability.' We will work on this to further improve our manuscript.

We included information on the selection of window width into our manuscript (see Page 5, Line 11-13).

**2.3 Comments on Results and Discussion**

**2.3.1 Page 5, Line 9**

**Are you really plotting discharge with lowflow on the top of the y-axis and peak discharge on the bottom-part of the y-axis?!?!? This is absolutely contra intuitive! Change! Furthermore, I would plot the dry season in red and the wet one in blue .....**

Indeed. Thank you for pointing at this. Reversing the y-axis/colors makes the understanding of our figures way more intuitive. We tried already and it looks better now. We will change our figures accordingly.

We changed color scales, plot set-up and method illustrations, so that high discharges and increases in discharge are displayed with blue colors. With regard to temperature, high/increasing values are still displayed with orange/red colors.

**2.3.2 Page 5, Line 14**

**This has to do with size. For such large rivers is in my opinion quite challenging to attribute a distinct regime characteristic. I think that classic regime classifications (nival, glacial, pluvial) is something you can attribute to mesoscale basins (up to 1000 km2 or so).**

Yes, the larger basins are, the more difficult it usually gets. We will rethink our regime descriptions and try to formulate a more suitable characterization.

We rephrased the paragraph on the seasonality of river runoff to move away the focus from the regime classification (Page 5, Line 25-30). It indeed is getting more difficult with larger catchment size. However, we have troubles to entirely abandon the distinction of flow regimes into nival, pluvial and complex. We hope that the Pardé-Coefficients and the raster hydrograph given in the appendix (Fig. A1 and Fig. F1) help to apprehend our approach. Furthermore, we now hint at detailed regime descriptions for Switzerland (Page 7, Line 17-18).

**2.3.3 Page 5, Line 23**

**I see it, very nice!**

Cool.

**2.4 Page 6, Line 1-6**

**Have you tought to createa proxy for liquid/solid precipitation and combining P and T? For the Rhine in Basel an additional station at elevation > 1000 m might be useful (Davos?)**

We did not think of creating a liquid/solid precipitation proxy yet. It indeed might provide very interesting supporting information for our discussion. We will look into this. We did not include station Davos into our study as it did not fulfil our criterion that time series should not have data gaps longer than 60 days. But we will look at the time series again and try. Maybe also stations Samedan (1709 m) or Chaumant (1136 m) turn out to be useful in this regard.

We tried different ways of calculating a liquid/solid proxy using available temperature and precipitation data, for station Davos, also for stations Samedan and Chaumant, which are part of the manuscript (Fig. D1). Our results hint at less solid precipitation in recent decades. However, as we would need to explain a new analytical tool and as this finding is not new and described in other studies very well already, we decided not to add an additional figure. We hope that refering the reader to other studies is enough for this aspect.

**2.4.1 Page 6, Line 19**

**Too few stations to make any speculation on that regard**

Yes, the limited number of stations available with such data length and quality limits the significance of our results. This might be the most vulnerable point of our analysis. However, consistent analysis results for stations and parameters investigated make us confident that even if the number of stations is limited, we attain meaningful results that are worth discussing. We recognize that we have to be very careful on what speculations to make and to clearly indicated what to be a robust finding of our analysis and what just speculation. We will work through our manuscript again to improve our writing in this regard.

We changed the sentence to better point at the speculative trait of this finding. Furthermore, we tried to make the reader more sensitive for the issue of limited number of data (see comment above and changes made in chapters 'Study area and Data' and 'Conclusion').

**2.4.2 Page 7, Line 25**

**Also [Marty et al., 2017]**

Thank you. We will include it into our manuscript.

Included Page 8, Line 10.

**2.4.3 Page 8, Line 25**

**See, in German**

`https://hydrologischeratlas.ch/produkte/druckausgabe/fliessgewasser-und-seen/tafel-5-3` Fig. 1 `https://hydrologischeratlas.ch/downloads/01/content/Tafel{_}53.pdf`

Very nice maps! We were not aware of their existence. This information helps us to better understand the influence of hydropower and lakes on river runoff.

We browsed through the different maps and tools and the Hydrological Atlas of Switzerland. It is really nice and helped/will help us with our analysis.

**2.4.4 Page 8, Line 31**

**See also [Bosshard et al., 2013]**

We will include information of the mentioned work into our manuscript.

Included Page 7, Line 10.

**3 Marked-up manuscript version**

Marked-up manuscript version produced using 'latexdiff' on the following pages. It compares HESSD discussion manuscript and the revised version.

[revised manuscript text omitted]

Nival flow regime | Decrease seasonality
↑ Winter  ↓ Summer | Gradual change
Entire time frame | ↑ Low  ↓ High
↑ Very high |
| c) Cologne | Complex flow
Pluvio-nival | ↑ Winter and spring
↓ Summer and autumn | No clear onset
Nival + pluvial pattern | ↑ All prob. levels
U-shape |
| d) Wuerzburg | Rain-fed
Pluvial flow regime | ↑ All seasons | No clear onset | ↑ All prob. levels
Gradual + U-shape |

[Figure]

**Figure A1.** Elevation distribution (raster cells in 500 m resolution calculated based on EU-DEM v.1.1 by the EU Copernicus Programme) and Pardé-coefficients (mean monthly discharge divided by the mean annual discharge) (Pardé, 1933; Spreafico and Weingartner, 2005) for investigated river basins Wasserburg, Basel, Koeln and Wuerzburg.

[Figure]

**Figure B1.** Changes in quantiles for individual seasons (Spring: March-May, Summer: June-August, Autumn: September-November and Winter: December-February) for discharge measured at gauges Wasserburg (a), Basel (b), Cologne (c) and Wuerzburg (d). Points on top of panels indicated days/probabilities with significant changes according to the Mann-Kendall trend test. Time frame investigated: 1869-2016.

[Figure]

**Figure C1.** Changes in quantiles for individual seasons (Spring: March-May, Summer: June-August, Autumn: September-November and Winter: December-February) for precipitation measured at stations Bern (a), Basel (b) and Zuerich (c). Points on top of panels indicated days/probabilities with significant changes according to the Mann-Kendall trend test. Time frame investigated: 1869-2016.

[Figure]

**Figure D1.** Onset and evolution of changes and changes in quantiles for temperature and precipitation measured at stations Sion (a), Samedan (b), Neuchatel (c), Lugano (d), Geneve (e) and Chamont (f). Points on top of panels indicated days/probabilities with significant changes according to the Mann-Kendall trend test. Time frame investigated: 1869-2016.

[Figure]

**Figure E1.** Cumulative storage volume of large storage lakes (active storage volume more than 0.3 hm³) in the High Rhine basin until gauge Basel. Figure is based on information presented in Wildenhahn and Klaholz (1996). Time frame displayed: 1900-1985.

[Figure]

**Figure F1.** Raster hydrograph for gauge Wasserburg. In recent decades, hydropeaking (weekly pattern) due to the operation of high-head storage hydropower stations imprinted. Time frame displayed: 1869-2016.

**Table A1.** Additional climate stations investigated: station name, location (WSG 84), altitude [m], daily resolution time series investigated with temperature (T) and precipitation (P) and data source Federal Office of Meteorology and Climatology of Switzerland (MeteoSwiss).

| Station | Lat. | Lon. | Alt. | Vari. | Data source |
|---|---|---|---|---|---|
| Sion | 46.2186 | 7.3303 | 482 | T-P | MeteoSwiss |
| Samedan | 46.5264 | 9.8789 | 1709 | T-P | MeteoSwiss |
| Neuchatel | 47.0000 | 6.9533 | 485 | T-P | MeteoSwiss |
| Lugano | 46.0042 | 8.9602 | 273 | T-P | MeteoSwiss |
| Geneve / Cointrin | 46.2475 | 6.1278 | 411 | T-P | MeteoSwiss |
| Chaumont | 47.0492 | 6.9789 | 1136 | T-P | MeteoSwiss |

**Table B1.** Abbreviations and acronyms in alphabetical order.

| Abbreviation/Acronym | Description |
| --- | --- |
| AMO | Atlantic Multidecadal Oscillation |
| CEEMDAN | Complete Ensemble Empirical Mode Decomposition with Additive Noise |
| DOY | Day of the Year |
| EMD | Empirical Mode Decomposition |
| EEMD | Ensemble Empirical Mode Decomposition |
| GRDC | Global Runoff Data Center |
| IMF | Intrinsic Mode Function |
| MeteoSwiss | Federal Office of Meteorology and Climatology of Switzerland |
| MK | Mann-Kendall trend test |
| NAO | North Atlantic Oscillation |
| QDAY | Quantiles estimated on a daily basis |
| QMOV | Quantiles estimated within moving window |
| QYEA | Quantiles estimated on an annual level |
| RoS | Rain-on-Snow |
| TST | Theil-Sen trend estimator |